# A Review of Maternal Nutrition during Pregnancy and Impact on the Offspring through Development: Evidence from Animal Models of Over- and Undernutrition

**DOI:** 10.3390/ijerph17186926

**Published:** 2020-09-22

**Authors:** John F. Odhiambo, Christopher L. Pankey, Adel B. Ghnenis, Stephen P. Ford

**Affiliations:** 1Division of Agricultural Sciences, Florida A&M University, Tallahassee, FL 32307, USA; 2Formerly, Department of Animal Science, University of Wyoming, Laramie, WY 82071, USA; cpankey@osteo.wvsom.edu (C.L.P.); ghnenis@tamu.edu (A.B.G.); spford@wvu.edu (S.P.F.); 3Department of Biomedical Sciences, West Virginia School of Osteopathic Medicine, Lewisburg, WV 24901, USA; 4Department of Neuroscience and Experimental Therapeutics, Texas A&M University, Bryan, TX 77807, USA

**Keywords:** fetal programming, maternal obesity, maternal undernutrition, metabolic syndrome, epigenetics

## Abstract

Similarities in offspring phenotype due to maternal under- or over-nutrition during gestation have been observed in studies conducted at University of Wyoming. In these studies, ewes were either nutrient-restricted (NR) from early to mid-gestation, or fed an obesogenic diet (MO) from preconception through term. Offspring necropsies occurred at mid-gestation, late-gestation, and after parturition. At mid gestation, body weights of NR fetuses were ~30% lighter than controls, whereas MO fetuses were ~30% heavier than those of controls. At birth, lambs born to NR, MO, and control ewes exhibited similar weights. This was a consequence of accelerated fetal growth rates in NR ewes, and reduced fetal growth rates in MO ewes in late gestation, when compared to their respective controls. These fetal growth patterns resulted in remarkably similar effects of increased susceptibility to obesity, cardiovascular disease, and glucose intolerance in offspring programmed mostly during fetal stages of development. These data provide evidence that maternal under- and over-nutrition similarly induce the development of the same cadre of physical and metabolic problems in postnatal life.

## 1. Introduction

Accumulating evidence indicates that maternal and neonatal nutrition may have long lasting effects on the development of metabolic syndrome, which is characterized by insulin resistance, type II diabetes, obesity, dyslipidemia, hypertension, and cardiovascular disease [1]. Exposure to adverse nutritional environments during early life induces changes in tissue plasticity and may be critical in shaping how the body responds to metabolic challenges later in life [2,3]. There is clearly a link between maternal nutrition and offspring metabolic disorders; however, the underlying mechanisms are poorly understood. One of the purported mechanisms is modifications to the offspring epigenome [4]. Indeed, these early life exposures are known to influence expression of genes through epigenetic mechanisms (i.e., without changes in the primary DNA sequence). Ideally, epigenetic marks serve as a memory of early life exposures, and may potentially be copied from one cell generation to the next and perpetuate the risk for development of non-communicable diseases later in life or in future generations.

There are three distinct but closely interacting epigenetic mechanisms that can be influenced by nutritional factors. These include DNA methylation, histone modifications, and non-coding microRNAs, which together regulate the intensity and timing of gene expression during development [3,5]. Discovery of these epigenetic regulatory mechanisms have unveiled the limitations involved in genetic sequence analyses, and further reveal the importance of considering the broader picture, including all the complex mechanisms involved in protein expression [6]. It is important to consider that although epigenetic analyses will likely provide information describing the changes in genetic regulation that are induced by maternal nutrition, long term metabolic dysregulation as a result of epigenetic modifications may also provide feedback to additionally alter the epigenetic state, theoretically leading to additional epigenetic dysregulation [7,8].

## 2. Evidence of Impacts of Maternal Nutrition on Epigenetic Marks in the Progeny

Metabolic imprinting describes the adaptive metabolic processes of the developing fetus in response to nutritional challenges throughout gestation that can have lasting effects into adulthood [9]. Mechanisms of metabolic imprinting are believed to likely involve epigenetic gene regulation [10]. In addition, the developmental origins of the health and disease hypothesis propose that epigenetic mechanisms may influence developmental pathways leading to susceptibility to chronic disease [11].

Several animal models have been used to demonstrate the epigenetic effects of maternal nutrition on early fetal development. The earliest models included mice with metastable epialleles. The term “metastable” refers to the labile nature of the epigenetic state of these alleles, while “epiallele” defines their potential to maintain epigenetic marks over multiple generations [12]. These metastable epialleles are mammalian genes or genomic regions that are variably expressed, even in the absence of genetic heterogeneity, due to epigenetic modifications established during early development. Metastable epialleles are thought to be particularly vulnerable to environmental influences [13]. Several studies have demonstrated that gestational exposure to nutritional agents and other environmental factors alter epigenetic marks at metastable epialleles [12,14,15].

Two notable examples of these epialleles are the viable yellow agouti (*A^vy^*) [16,17] and axin-fused (*Axin^Fu^*) [18,19] alleles that have been well studied in the mouse. The *A^vy^* allele displays a range of coat colors from completely yellow to wild-type agouti [17]. A yellow coat correlates with a hypomethylated, thus active, epiallele. In the *A^vy^* mice, supplementation of methyl donors including folate during pregnancy increased DNA methylation of the agouti gene in the *A^vy^* offspring leading to a decrease in agouti expression and browning of the coat color [14,20]. Later on, Dolinoy et al. [12] demonstrated similar phenotypic changes in agouti mice following maternal supplementation with genistein during pregnancy. These phenotypic changes were attributed to increased methylation at the *A^vy^* locus of the agouti mice despite the fact that genistein is not a methyl donor. Interestingly, this genistein-induced hypermethylation protected the *A^vy^* offspring from obesity in adulthood. This observation provided a link between maternal nutrition and susceptibility to chronic disease in the *A^vy^* progeny.

The axin-fused (*Axin^Fu^*) mice have a spectrum of tail phenotypes, from severely kinked to completely normal. The extent of the kink in the tail is inversely related to the degree of methylation at the epiallele [21]. The mutant phenotype with severely kinked tails correlates with hypomethylated epiallele, whereas mice with silent, phenotypically normal tails are more hypermethylated within the *Axin^Fu^* locus. Similar to the *A^vy^* model, dietary methyl donor supplementation of female mice before and during pregnancy induced DNA hypermethylation, resulting in reduced tail kinking in the *Axin^Fu^/+* offspring [15]. Together, these studies demonstrate that early life exposures can be “stored” in the cellular memory epigenetically, and may foster developmental consequences.

## 3. Impact of Maternal Nutrition on Fetal Programming in Animal Models

Early epidemiological studies linked maternal nutrition with the etiology of chronic disease in offspring. More specifically, both maternal undernutrition [22,23] and maternal obesity [24,25,26] similarly alter fetal growth and development, which leads to a predisposition of offspring to chronic diseases in their adult life. Indeed, our studies with sheep models of maternal undernutrition and overnutrition similarly show increased offspring susceptibility to obesity, cardiovascular disease, and insulin/glucose dysregulation that is programmed during fetal stages of development [27,28,29]. The long-term consequences of fetal exposure to growth-restricted or obese gestational environments include increased prevalence of indicators for metabolic syndrome [30,31]. The etiology of these complications are seemingly related to abnormal organ and tissue development in utero due to insulin resistance, and secondary hyperinsulinemia induced by the suboptimal maternal diet during critical periods of development.

The “thrifty phenotype” hypothesis of Hales and Baker [32] proposes that undernutrition during early life results in the reallocation of energy and nutrients to favor development of organs critical for immediate survival (e.g., brain and heart) at the expense of organs and systems that are deemed less critical for immediate survival (e.g., pancreas and kidneys). This contributes to eventual dysfunction or failure of those ‘neglected’ systems in adulthood. Although an energetically thrifty phenotype may be beneficial in the face of food shortage, it constitutes a potentially dangerous strategy for individuals who in their adult life experience an abundance of high-energy foods, as many do in affluent societies [10]. With the recent rise in childhood obesity, maternal obesity and excessive gestational weight gain are now considered risk factors for adverse acute and chronic offspring outcomes through epigenetic mechanisms [23]. However, what remains perplexing then is: how can maternal over-nutrition produce the same collection of health outcomes as the thrifty phenotype?

This paradox about similarities in offspring phenotype due to fetal exposure to either maternal under- or over-nutrition during gestation can at best be answered by observations from studies conducted at University of Wyoming utilizing sheep models [27,28,29,33,34]. In these studies, nutrient restricted ewes (NR) received 50% of the nutritional requirements over the first half of gestation (d0–75; term = 150 days), followed by re-alimentation to 100% of nutritional requirements through term (d75–150), and obese ewes (MO) were fed 150% of nutritional requirements from 60 days prior to conception through term. Both nutritional models utilized the same pelleted diet, which controlled for micro and macro nutrient content. Fetuses were collected from some ewes of each group at mid (d75) and late gestation (d135), with the remaining ewes in each group were allowed to lamb. Lambs and fetuses of both NR and MO ewes exhibited remarkably similar phenotypes with respect to fetal and postnatal development, and alterations in glucose-insulin dynamics, as depicted in Table 1.

At mid gestation, NR fetuses weighed ~30% less than their respective controls, whereas MO fetuses weighed ~30% more than control fetuses. However, birthweight did not differ between NR, MO, or controls. This observation was a result of accelerated NR fetal growth in the second half of gestation, but a reduced MO fetal growth over this period when compared to their respective controls (Figure 1).

The accelerated growth of NR fetuses after re-alimentation to 100% of nutritional requirements at mid-gestation was attributed to significantly increased placental vascularity and nutrient transporter activity [35]. In contrast, MO fetuses suffered growth restriction over the second half of gestation, similarly attributed to decreased placental vascularity and nutrient transporter activity [38].

We can therefore conclude that fetuses of NR and MO ewes both experienced a period of nutrient deprivation in the first or second half of gestation, respectively. In either case, exposure to nutrient deprivation during fetal development leads to an adaptive reallocation of nutrients favoring the development of the organs most critical for sustaining life, and therefore the development of a thrifty phenotype. We have also observed decreased expression of β-cells in the fetal pancreas in both NR [39] and MO [40] fetuses, directly supporting this hypothesis (Table 1).

The embryo-fetal stage is also crucial for skeletal muscle development, and for adipose and connective tissue development. Considering that skeletal muscle is one of the largest organs in non-obese subjects and a major site of insulin- and exercise-stimulated glucose disposal, proper fetal skeletal muscle development is important for the metabolic health of the offspring. Our studies have shown that maternal nutrient restriction reduced the formation of secondary myofibers and the ratio of secondary to primary myofibers during fetal development [36], resulting in reduced offspring skeletal muscle mass and function [47]. Further, maternal over-nutrition during gestation resulted in impaired myogenesis by elevating intramuscular adipogenesis and fibrogenesis, increasing intramuscular fat and connective tissue content in offspring muscle [48,49,50]. A shift from myogenesis to adipogenesis or fibrogenesis will replace muscle fibers with adipocytes or fibrous tissues, impairing the physiological functions of skeletal muscle, such as the reduction in muscle force and oxidative capacity. In addition, enhanced adipogenesis within muscle leads to skeletal muscle insulin resistance, which plays a key role in the pathogenesis of type 2 diabetes.

Another likely mechanism for the similar phenotypic programming in both nutritional models is hypercortisolemia [51]. Recent data from our lab showed that exposure to glucocorticoid levels higher than appropriate for current developmental stages induced offspring metabolic dysfunction [52,53]. Although the placenta serves to protect the fetus from maternal cortisol [54], the inert cortisone that enters the fetal compartment can serve as a source for fetal cortisol production via increased enzymatic metabolism in the periphery. Specifically, in the fetal liver and adipose tissue, increased expression of 11β-hydroxysteroid dehydrogenase enzymes catalyze the conversion of cortisone to cortisol, thereby providing a mechanism for an ACTH-independent increase in circulating fetal cortisol in MO fetuses [53]. In response to unfavorable nutritional environments, increased cortisol production combats the low energy status of the fetus during the period of nutrient deprivation described above. In addition to its role in glucose metabolism however, cortisol can also differentiate cells and impact developing tissues, therefore leading to the dysregulated development and function of tissues and organ systems when inappropriately circulating at elevated levels. For example, hypercortisolemia in dietary programming models has been associated with impaired pancreatic beta cell development and function [27], neural development associated with appetite regulation [34], increased adiposity [55], and impaired cardiovascular development [56]. Tying this back to our sheep models, we observed similar findings in both undernourished [29,50] and over-nourished [43,57,58] fetuses, particularly with regard to alterations in cardiac morphometry and function. Through these various mechanisms, elevated cortisol during development predisposes offspring to the development of multiple indications of metabolic syndrome.

Expanding on the previous studies, we have also shown that many indications of the metabolic syndrome are observed in second [55,59] and third (unpublished data) generation offspring as well. In these subsequent generations, indications of metabolic syndrome deteriorate from generation to generation and are latent until individuals are exposed to metabolic stressors such as ad libitum feeding, bolus glucose administration, or pregnancy. Together, these findings suggest that metabolic imprinting from maternal nutritional status can be passed through the germ line, but the effects of maternal diet can be ameliorated with a healthy lifestyle.

## 4. Summary and Implications for Human Health Disparities

The data reviewed above provide robust evidence that maternal malnutrition alters the phenotype of both prenatal and postnatal offspring. These data further support the concept that both maternal overnutrition and maternal undernutrition similarly alter offspring phenotype, resulting in predisposition to metabolic syndrome later in development. The observed phenotypic differences in the offspring are likely derived from epigenetic responses to the altered nutritional state during gestation, subsequent alterations in gene expression, and ultimately, the development of a thrifty phenotype. In both undernourished and over-nourished pregnancies, the thrifty phenotype develops in response to a period of reduced fetal growth, which is disadvantageous in the face of nutrient availability, or even excess nutrient availability postnatally (Figure 2).

Although undernutrition during pregnancy remains a significant public health issue, there is rising concern regarding the adverse effects of over-nutrition during pregnancy. Rates of overweight and obese women entering pregnancy are continuously increasing [60], which is concerning as we develop a more complete picture of the detrimental consequences for the mother and her baby. With consideration of the literature reviewed above, it is not unreasonable to suspect that overweight and obese pregnancies may epigenetically affect the developing fetus and predispose them to the development of chronic disease.

## Figures and Tables

**Figure 1 ijerph-17-06926-f001:**
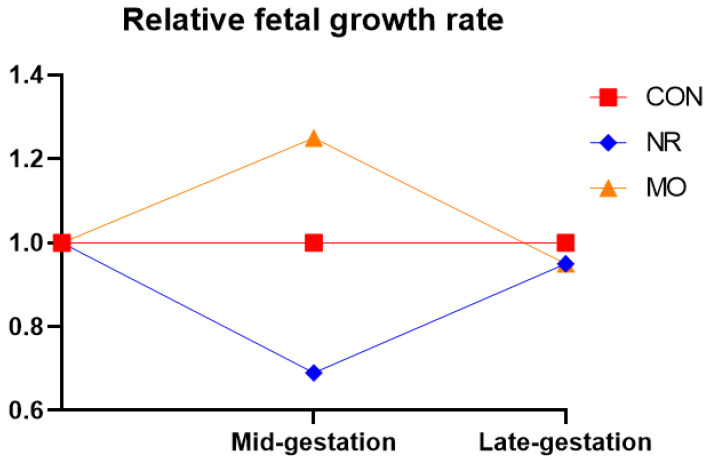
Relative fetal growth rate at mid-gestation (day 75, term = 150 d) and late gestation (day 135) in fetuses from nutrient restricted (NR), obese (MO), and control (CON) ewes [27,29,35,37,38,44].

**Figure 2 ijerph-17-06926-f002:**
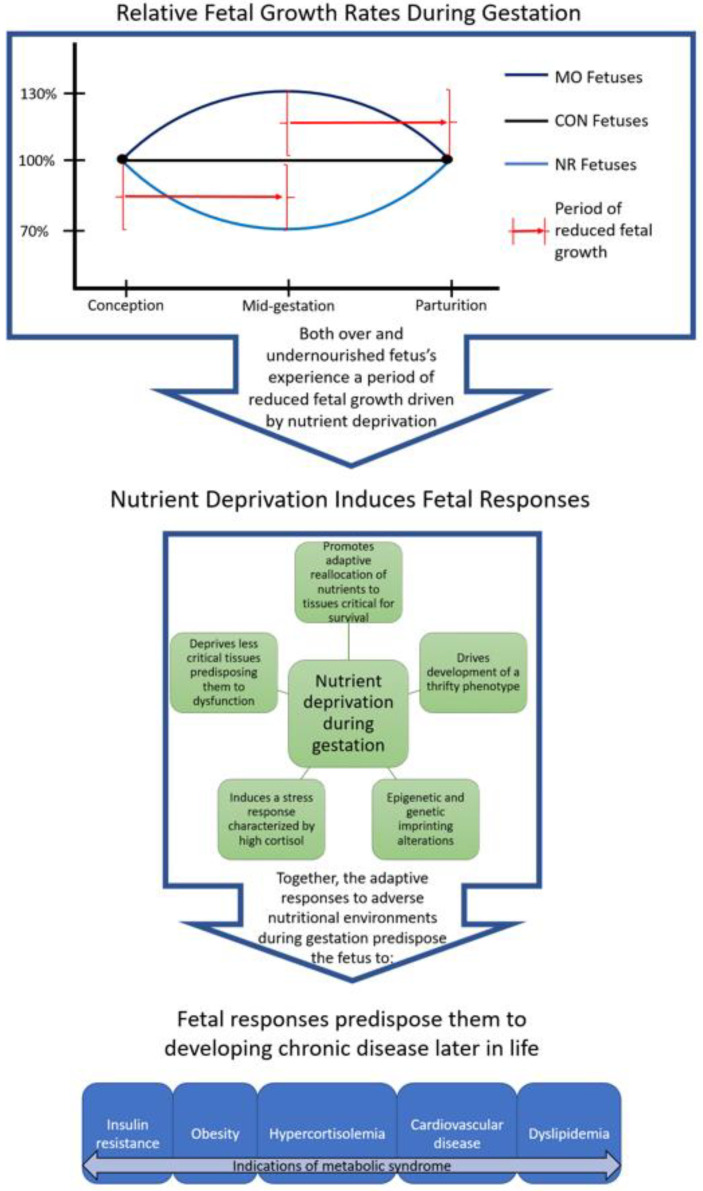
Summary diagram of the mechanisms behind similar fetal outcomes from maternal obesity (MO) and nutrient restriction (NR) relative to controls (CON). In both circumstances, there is a period of fetal growth restriction resulting from nutrient deprivation. Nutrient deprivation of tissues and organs that are not immediately critical for survival, but still play important metabolic roles, predisposes the fetus to developing metabolic derangements in postnatal life. Predisposition to metabolic disease can persist through multiple generations but is often latent until the individual is exposed to a metabolic stress [27,29,35,37,38,44,59].

**Table 1 ijerph-17-06926-t001:** Comparison of fetal and adult offspring characteristics of nutrient restricted (NR) and overfed-obese (MO) to ewes fed to requirements.

Mid−Gestation (Day 78) Fetus	NR Offspring	MO Offspring	Reference(s)
− fetal weight	−	+	[29,35,36,37],[37,38]
− crown rump length	−	+	[29,35,37],[37,38]
− liver wt	+	+	[29],[27]
− pancreatic β−cell numbers	+	+	[39],[27,40]
− cardiac ventricular wt/fetal wt	+	+	[41],[37]
− plasma glucose	−	+	[29],[27]
− plasma insulin	ND	+	[N/A],[27]
− plasma cortisol	+	+	[37],[27,37]
− plasma cholesterol	+	+	[35],[42]
− plasma triglycerides	−	+	[35],[42]
**Late−gestation (Day 135) fetus**			
− fetal wt	ND	ND	[38],[38]
− crown rump length	ND	ND	[38],[38]
− liver wt	ND	ND	[41],[40]
− pancreatic β−cell numbers	−	−	[39],[40]
− cardiac ventricular wt./fetal wt	+	+	[41],[43]
− plasma glucose		ND	[N/A],[40]
− plasma insulin		ND	[N/A],[40]
− plasma cortisol		ND	[N/A],[40]
− plasma cholesterol	ND	+	[35],[42]
− plasma triglycerides	−	+	[35],[42]
**Newborn lambs**			
− birth weight	ND	ND	[41],[27,44]
− crown rump length		ND	[N/A],[27]
− plasma glucose	+	+	[34],[27]
− plasma insulin	−	−	[34],[44]
− plasma cortisol	+	+	[34],[44]
− plasma leptin	−	−	[34],[44]
**Adult offspring**			
− appetite	+	+	[45],[46]
− growth rate	+	+	[33],[46]
− insulin resistance	+	+	[33,45],[46]
− plasma leptin	+	+	[45],[46]
− adiposity to ad. lib. feeding	+	+	[45],[46]
− left ventricular wall thickness	+	+	[45],[46]

“+” indicates an increase (*p* < 0.05) and “−” indicates a decrease (*p* < 0.05) relative to controls, while “ND” indicates no difference. References listed in the order: [NR offspring], [MO offspring].

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
