# Peer review of "A Review of Maternal Nutrition during Pregnancy and Impact on the Offspring through Development: Evidence from Animal Models of Over- and Undernutrition"

_ijerph, 2020, doi:10.3390/ijerph17186926_

Round 1

Reviewer 1 Report

Congratulations for recognizing and honoring, your senior investigator S.P. Ford post humus.

The effect of maternal over and/or under nutrition in offspring phenotype is very important and as you pointed out it is critical in the current world situation. Since you are focusing this review on the similarities in offspring phenotypes I believe it would be more appropriate for you to change the name of the paper to reflect how the difference in maternal diet during pregnancy could affect the weight of the fetus at birth. I would suggest you expand your review concentrating on the effects of maternal diet and how it can affect the offspring weigh at birth.

Author Response

Reviewers Comments

Comments and Suggestions for Authors from Reviewer #1

Congratulations for recognizing and honoring, your senior investigator S.P. Ford post humus.

The effect of maternal over and/or under nutrition in offspring phenotype is very important and as you pointed out it is critical in the current world situation. Since you are focusing this review on the similarities in offspring phenotypes I believe it would be more appropriate for you to change the name of the paper to reflect how the difference in maternal diet during pregnancy could affect the weight of the fetus at birth. I would suggest you expand your review concentrating on the effects of maternal diet and how it can affect the offspring weigh at birth.

            Thank you for your congratulations, and for your insightful comments. We have changed the title to more specifically address the models we discuss. However, since weight is only one aspect of multifactorial programming outcomes, we did not incorporate the weight specific changes suggested above. Additionally, weight is often an indirect measurement of adiposity, and does not incorporate differences in fat mass, fat free mass, etc., which are more accurate predictors of future health disparities given the roles they play in metabolism. Finally, reviewer 2 also made insightful comments regarding the title, and we wanted to incorporate them as well.

Reviewer 2 Report

In this manuscript, Odhiambo et al. summarize the results from different published reports on the effects of maternal over-nutrition versus under-nutrition during gestation and in the offspring phenotype. They also discuss the potential involvement of epigenetic modifications. While many of the cited results are referred to gestational nutrition of animal models, the authors also briefly discuss the implications for human health.

General comment:

This review is straightforward and properly addresses the intriguing and longstanding paradox on how maternal over- or under-nutrition can lead to similar offspring phenotypes. The proposed mechanisms involved are interesting. Overall the paper is well written and the cited results, published over the last decades, are convincing. However, some concerns should be addressed before publication.

Specific comments

  1. Page 1, Title. Most of the experimental data in this review are referred to animal models, something that must be indicated the in the title.
  2. Page 1, Title. “….and implications for human disparities”. The relevance of the results derived from animal models in humans is discussed in a very short item (#4). Therefore it would be more appropriate to skip the human implications from the title.
  3. Table 1. Different fetal and adult offspring characteristics are listed. According to the table legend, these parameters are derived from references 30, 39-44. However the table would be much more informative if each experimental parameter is associated with the specific reference(s) in an additional column.
  4. Figure 1. The authors must indicate the original source (Reference/s) of the data represented in this figure. Also, the statistical significance of the results should be provided.
  5. The authors may wish to enrich the review by including a hypothetical figure/scheme summarizing the results and the potential mechanisms involved.

Minor comment

  1. Page 6, lines 210-212. While in line 210 it is stated that “This research received no external funding”, in lines 211 and 212, it is stated that “Authors acknowledge funding support from National Institutes of Health through the following grants: NIH HD070096-01A1 and WYO INBRE P20RR16474”. This is somehow confusing and the authors may wish to clarify this point.

Author Response

Comments and Suggestions for Authors from Reviewer #2

In this manuscript, Odhiambo et al. summarize the results from different published reports on the effects of maternal over-nutrition versus under-nutrition during gestation and in the offspring phenotype. They also discuss the potential involvement of epigenetic modifications. While many of the cited results are referred to gestational nutrition of animal models, the authors also briefly discuss the implications for human health.

General comment:

This review is straightforward and properly addresses the intriguing and longstanding paradox on how maternal over- or under-nutrition can lead to similar offspring phenotypes. The proposed mechanisms involved are interesting. Overall the paper is well written and the cited results, published over the last decades, are convincing. However, some concerns should be addressed before publication.

Specific comments

  1. Page 1, Title. Most of the experimental data in this review are referred to animal models, something that must be indicated the in the title.

Thank you for your comment, we have edited the title (line 3) to address this concern.

  1. Page 1, Title. “….and implications for human disparities”. The relevance of the results derived from animal models in humans is discussed in a very short item (#4). Therefore it would be more appropriate to skip the human implications from the title.

Thank you for your comment, we have edited the title to address this concern.

  1. Table 1. Different fetal and adult offspring characteristics are listed. According to the table legend, these parameters are derived from references 30, 39-44. However the table would be much more informative if each experimental parameter is associated with the specific reference(s) in an additional column.

Thank you for your comment, Table 1 has been edited to include this information (lines 123-125)

  1. Figure 1. The authors must indicate the original source (Reference/s) of the data represented in this figure. Also, the statistical significance of the results should be provided.

Thank you for your comment, this revision significantly strengthened our work. Table 1 has been edited to include this information (lines 123-124), and we have added additional references. For statistical significance, we note in the legend that significant p values are less that 0.05. We have also added additional references to this table to strengthen this summary considering this concern. We have not added specific p-values to each finding, however. This is due to the clutter it would cause within the figure as some findings are supported by multiple studies, and therefore have multiple p-values. All findings summarized within table one are significant with alpha set at the conventional 0.05.

  1. The authors may wish to enrich the review by including a hypothetical figure/scheme summarizing the results and the potential mechanisms involved.

Thank you for your comment. Once again, this revision has greatly strengthened this paper. We have included this new figure (fig 2.) on lines 198-206, and referenced it on line 196.

Minor comment

  1. Page 6, lines 220-212. While in line 210 it is stated that “This research received no external funding”, in lines 211 and 212, it is stated that “Authors acknowledge funding support from National Institutes of Health through the following grants: NIH HD070096-01A1 and WYO INBRE P20RR16474”. This is somehow confusing and the authors may wish to clarify this point.

Thank you for this comment. We have revised the statements on lines 222-225 to better state our funding, and acknowledgments.

Reviewer 3 Report

As a review paper, the present study focus on the maternal over-nutrition versus under nutrition: epigenetic impact on the fetus and implications for human health disparities. However, the present study only focus on some maternal epigenetic phenomenon, but the potential mechanisms was rarely reported. Meanwhile, there were a large amount of novel paper focused on the maternal nutrition, but the references that authous cited were not novel enough. 

Moreover, the over nutrition and undr nutrition could contain several nutrients' effects, and their roles were not the same. 

in conclusion, the present study should added these discussion. 

Author Response

Comments and Suggestions for Authors from Reviewer # 3

As a review paper, the present study focus on the maternal over-nutrition versus under nutrition: epigenetic impact on the fetus and implications for human health disparities. However, the present study only focus on some maternal epigenetic phenomenon, but the potential mechanisms was rarely reported. Meanwhile, there were a large amount of novel paper focused on the maternal nutrition, but the references that authous cited were not novel enough. 

Moreover, the over nutrition and undr nutrition could contain several nutrients' effects, and their roles were not the same. 

in conclusion, the present study should added these discussion. 

Thank you for reviewing our paper, and for your insightful comments. While epigenetics are indeed an important mechanism, you’re correct that we did not focus extensively on these mechanisms. Our goal was to introduce potential mechanisms involved in our observations that we could refer to throughout the text. Epigenetic mechanisms of fetal programming are one of multiple mechanisms implicated in fetal programming, and that subject could encompass an entire review alone. This was not the intended direction of this review, and we may have misled you to believe that was the case with our original title. We have revised the title, and hopefully will not mislead others regarding what we’ve reviewed here.

We have also added additional references but disagree with the “novelty” of our references. A primary goal of this review was to summarize over a decade of research that was accomplished in the ford lab. This was a strategic approach which allows for ease of comparisons between two nutritional models within the same model organism. Many of these studies were done at the Center for the Study of Fetal Programming, further reducing variability in study design, research and lab procedures, etc., which further allows for ease of comparison between studies, and greatly strengthens the conclusions and findings displayed in this paper. This also directly addresses your second comment regarding the potential effects of individual nutrients. For all studies from our group, we used the same standardized, pelleted diet (details on the nutrition content can be found at doi: 10.1016/j.jnutbio.2007.06.003). The only difference between our treatment groups was the volume of this standardized diet. Therefore, all individual nutrients (both macro and micro) were controlled in our studies, removing that potential confounding variable and greatly strengthening this review. We felt this was an important detail to include in the text, and therefore added “Both nutritional models utilized the same pelleted diet which controlled for micro and macro nutrient content.” To lines 119-120.

Round 2

Reviewer 1 Report

It is good to see that you were able to include a summary diagram of the mechanism of the effects of maternal nutrition during fetal development. I think this improves the quality of the paper.

Author Response

Thanks for your favorable review. We believe that the concerns raised about the introduction of the manuscript has were adequately responded to. Section 1 and 2 of the manuscript is a continuous review of literature that discusses the mechanisms by which the maternal environment impacts the offspring epigenome leading to the maladies being described here (metabolic syndrome). We have adequately referenced those studies and introduction leads seamlessly to our studies in Wyoming, which is the focus of the review. Thanks again for your service.

Reviewer 2 Report

The authors have properly addressed most of my concerns.

Minor comment: Table 1, last column on the right now includes de requested references. However, the meaning of “[N/A]” should be clarified in table legend.

Author Response

Thank you for your kind review. The manuscript looks much better now.

Reviewer 3 Report

accept 

Author Response

Thank you too for your kind review, We have done all the spell checks and corrected the misspelled words. The manuscript looks a lot better after incorporating your suggestions. Thanks.

This manuscript is a resubmission of an earlier submission. The following is a list of the peer review reports and author responses from that submission.